# Daptomycin treatment impacts resistance in off-target populations of vancomycin-resistant *Enterococcus faecium*

**Clare L. Kinnear**[1]*, **Elsa Hansen**[2], **Valerie J. Morley**[2], **Kevin C. Tracy**[1], **Meghan Forstchen**[1], **Andrew F. Read**[2,3], **Robert J. Woods**[1]

**1** Department of Internal Medicine, Division of Infectious Diseases, University of Michigan, Ann Arbor, Michigan, United States of America, **2** Center for Infectious Disease Dynamics and Department of Biology, Pennsylvania State University, University Park, Pennsylvania, United States of America, **3** Huck Institutes of the Life Sciences and Department of Entomology, Pennsylvania State University, University Park, Pennsylvania, United States of America

* clarelkinnear@gmail.com

**Data Availability Statement:** Sequencing data is available from NCBI Genebank (accession number

## Abstract

The antimicrobial resistance crisis has persisted despite broad attempts at intervention. It has been proposed that an important driver of resistance is selection imposed on bacterial populations that are not the intended target of antimicrobial therapy. But to date, there has been limited quantitative measure of the mean and variance of resistance following antibiotic exposure. Here we focus on the important nosocomial pathogen *Enterococcus faecium* in a hospital system where resistance to daptomycin is evolving despite standard interventions. We hypothesized that the intravenous use of daptomycin generates off-target selection for resistance in transmissible gastrointestinal (carriage) populations of *E. faecium*. We performed a cohort study in which the daptomycin resistance of *E. faecium* isolated from rectal swabs from daptomycin-exposed patients was compared to a control group of patients exposed to linezolid, a drug with similar indications. In the daptomycin-exposed group, daptomycin resistance of *E. faecium* from the off-target population was on average 50% higher than resistance in the control group (*n* = 428 clones from 22 patients). There was also greater phenotypic diversity in daptomycin resistance within daptomycin-exposed patients. In patients where multiple samples over time were available, a wide variability in temporal dynamics were observed, from long-term maintenance of resistance to rapid return to sensitivity after daptomycin treatment stopped. Sequencing of isolates from a subset of patients supports the argument that selection occurs within patients. Our results demonstrate that off-target gastrointestinal populations rapidly respond to intravenous antibiotic exposure. Focusing on the off-target evolutionary dynamics may offer novel avenues to slow the spread of antibiotic resistance.

## Introduction

Antimicrobial resistance emerges and spreads in response to antimicrobial treatment [1]. For the microbial population being intentionally targeted by drug treatment, selective pressure

PRJNA673360) All other data is available in supplementary information.

**Funding:** This research was supported by funding to AFR from Eberly College of Science and Huck Institutes of the Life Sciences, Pennsylvania State University and to RJW from the National Institutes of Health (R01AI143852 and K08 AI119182). The funders had no role in study design, data collection and analysis, decision to publish, or preparation of the manuscript.

**Competing interests:** I have read the journal's policy and the authors of this manuscript have the following competing interests: A.F.R reports consulting with Foamix Inc and unrestricted grants from Merck, outside the submitted work. All other authors declare that no competing interests exist.

**Abbreviations:** AIC, Akaike Information Criterion; BMD, broth microdilution; BSI, blood stream infection; BWA, Burrows-Wheeler Aligner; ClsA, cardiolipin synthase A; DIC, Deviance Information Criterion; GATK, Genome Analysis Toolkit; $MIC_c$, computed minimum inhibitory concentration; OD, optical density.

favoring resistance is an unavoidable consequence of suppressing population growth. However, antimicrobial exposure is not limited to the site of infection and so can exert selective pressure on so-called off-target or bystander populations [2–4]. This off-target selective pressure is particularly problematic for colonizing opportunistic pathogens, which include the multidrug-resistant pathogens of greatest concern [5]. Antimicrobial pressure imposed on these colonizing organisms has no therapeutic benefits, but can select for antibiotic resistant infections in the treated patient, or for the transmission of resistant isolates to other patients [6]. Increases in resistance in gastrointestinal carriage populations following antimicrobial treatment has been demonstrated in multiple opportunistic pathogens including Enterobacteriaceae [7–12], *Enterococcus* [13], and Bacteroides [14]. In many situations, selection on off-target populations may be a major contributor to population-level resistance. For example, it has been estimated that as much as 90% of drug exposure experienced by *Klebsiella* is off-target [15]. What remains unclear is the evolutionary dynamics in these off-target populations. Whether changes in resistance result from colonization with more resistant isolates, up selection of low-density resistant isolates or de novo mutation is unknown, and studies exploring population diversity are lacking. This fundamental knowledge gap limits the ability to evaluate novel resistance management strategies in the off-target population such as choice of antimicrobial spectrum, use of antibiotic combinations, optimized routes of administration, and addition of novel adjuvant therapies [6].

The evolution of daptomycin resistance among vancomycin-resistant *Enterococcus faecium* (VR *E. faecium*) is a relevant and tractable system to study off-target selection. VR *E. faecium* is an important cause of hospital-acquired infections [16]. It spends the bulk of its life history asymptomatically colonizing the gastrointestinal tract of its host, but colonization is a key risk factor for clinical infections [17]. Intrinsic and acquired multidrug resistance are common in VR *E. faecium*, leaving daptomycin as one of the few remaining treatment options. Daptomycin is delivered exclusively as an intravenous formulation, with 6% of the drug being excreted in the feces [18], allowing the potential for off-target selection. Daptomycin resistance has been shown to arise within patients during treatment [19]. Additionally, a number of studies have reported prior daptomycin treatment as a key risk factor for infection with daptomycin-resistant VR *E. faecium* [20–22], which is consistent with off-target selection. Finally, transmission of VR *E. faecium* is common in hospital settings, meaning resistance that arises in one patient may pose a threat to others. That asymptomatic carriage can lead to symptomatic infection, and the fecal-oral route of infection indicate that off-target selection in the gastrointestinal tract is relevant to resistance threat to both the individual host and to others. We hypothesize that daptomycin exposure is associated with daptomycin resistance in the off-target population, which has not been demonstrated previously. If this hypothesis is true, it raises additional questions: Is there preexisting variation within patients prior to exposure on which selection can act; is there more variability within some patients than others; does daptomycin exposure result in a single resistance phenotype dominating the gut; and finally, is variation in resistance maintained over time?

To address this hypothesis and the resulting questions, we focus on an institution where daptomycin resistance in *E. faecium* has been observed to evolve within patients and across the whole hospital population [19,23]. We investigate the impact of daptomycin exposure on daptomycin resistance in *E. faecium* colonizing the intestinal tract by utilizing rectal swabs available from a prospective surveillance program. The organisms colonizing the intestinal tract are not the target of treatment, thus resistance in this population represents unintended, off-target evolution. We perform a cohort study, in which patients exposed to daptomycin are compared to patients exposed to a drug with similar indications, linezolid. The available samples allow us to isolate and measure resistance in multiple independent *E. faecium* colonies per patient swab

sample. We quantify the impact of drug exposure on the mean and the distribution of phenotypes in the colonizing population, which gives unique insight into the potential mechanism of competition and transmission in this pathogen. Finally, where samples are available, we explore changes in resistance over time within patients.

## Results

During the calendar year 2016, 6,726 patients had a rectal swab to screen for VRE colonization; of these, 618 patients were positive for VRE. Treatment with daptomycin within this pool of patients was low (23/618 patients, Fig 1), partially due to a change in antibiotic use in 2015 away from daptomycin [23]. Fourteen patients met the inclusion criteria for the daptomycin exposure cohort with at least 3 doses of daptomycin therapy in the 6 months prior to the positive swab (daptomycin group). A further 15 patients met the criteria for the control cohort, with no known prior daptomycin treatment, and at least 6 doses of linezolid therapy daily as linezolid is dosed twice daily, whereas daptomycin is generally dosed once (Fig 1, see Methods for further details). The first sample from each patient to meet these criteria is considered the index sample for that patient.

Multiple *E. faecium* clones were isolated from each VRE-positive index sample using Enterococcosel agar. While the inclusion criteria required a positive result for VR *E. faecium*, the collection protocol resulted in isolation of both vancomycin-resistant and vancomycin-susceptible *E. faecium*. For 2 of the daptomycin-treated samples and 5 of the control samples,

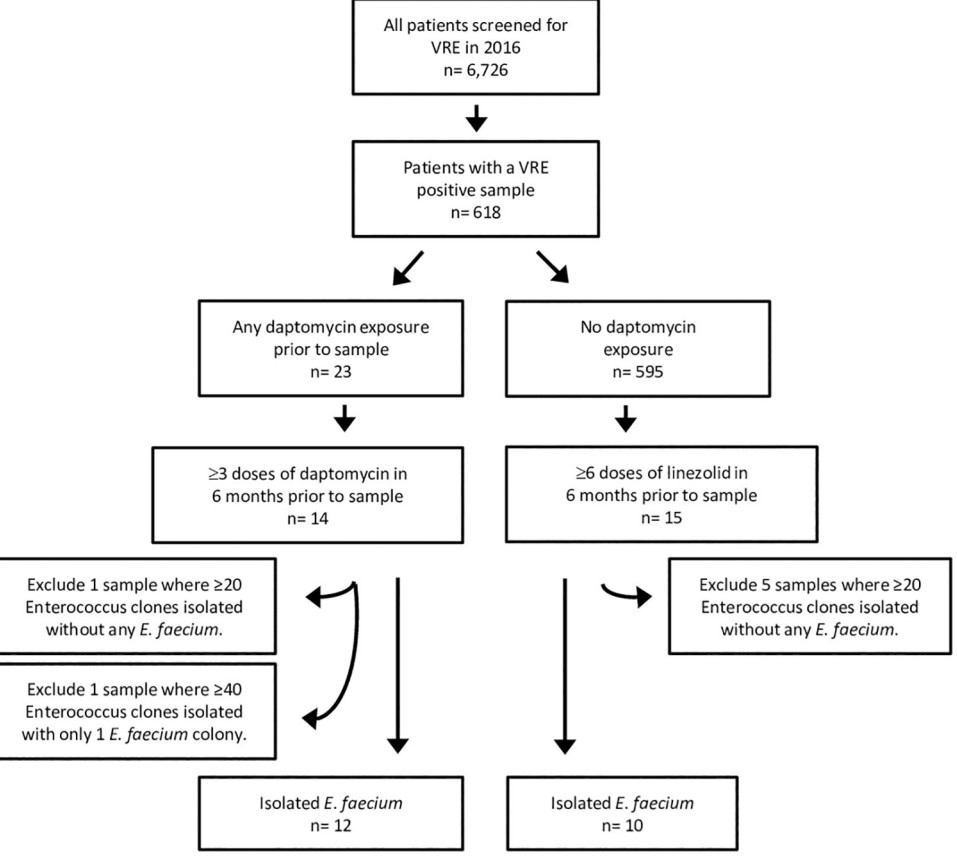

**Fig 1. Identification of patients meeting the daptomycin exposure and control study definitions.**

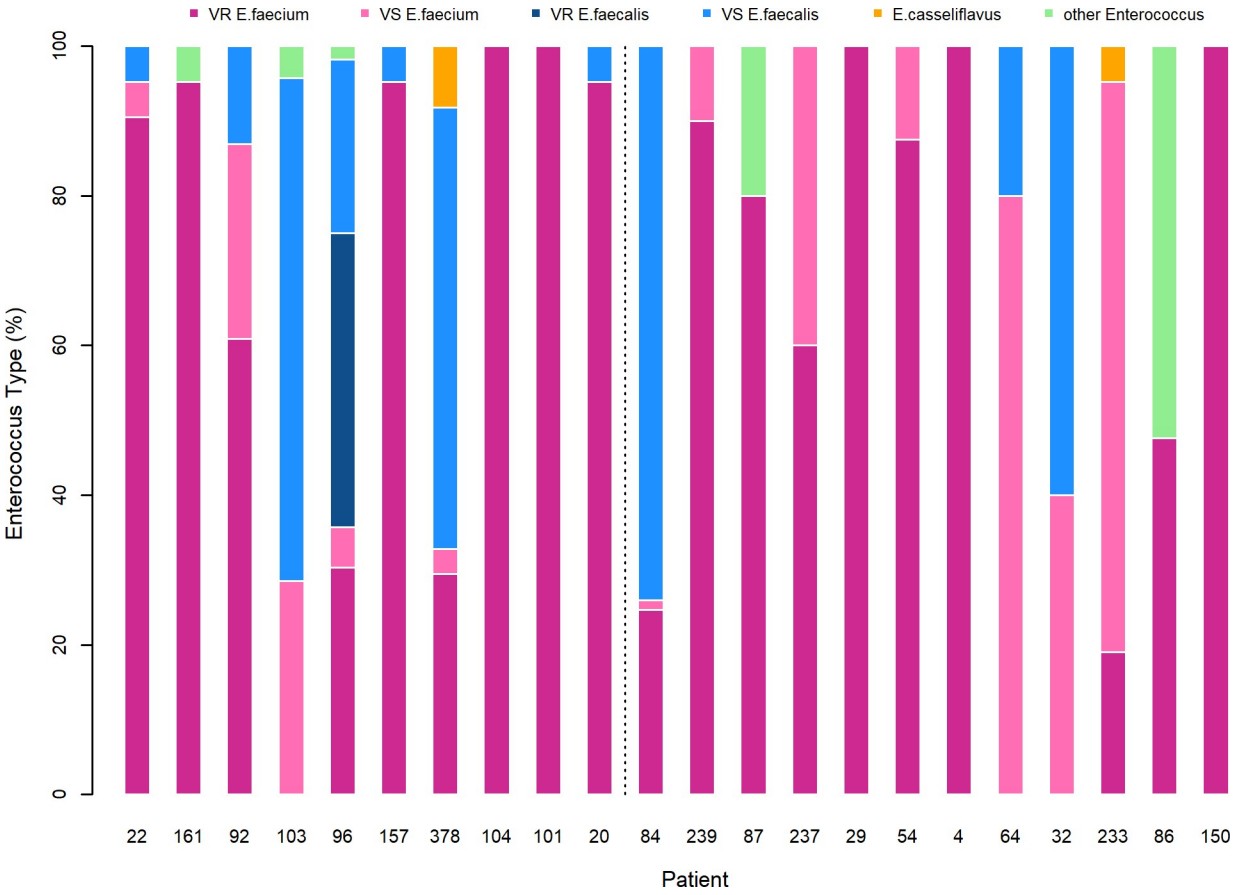

**Fig 2. Proportion of different *Enterococcus* types within each patient sample.** Each bar represents all the isolates from the patient's index sample divided by the percentage of isolates that were identified as *E. faecium*, *E. faecalis*, *E. casseliflavus*, *E. gallinarum*, or other *Enterococcus*. Other *Enterococcus* is used for clones that were morphologically *Enterococcus* but did not contain any of the sequences used for PCR identification. Patients to the left of the dotted line are in the control cohort and to the right are in the daptomycin-treated cohort. The underlying data for this figure can be found in S1 Data.

*E. faecium* was either absent or at a density (relative to other *Enterococcus* species) that did not allow isolation of sufficient *E. faecium* colonies. Samples where no *E. faecium* was isolated after sampling 20 random *Enterococcus* colonies (6 patient samples), or where only 1 colony was isolated after sampling 40 random *Enterococcus* colonies (1 patient sample), were excluded from further analysis (Fig 1). For the remaining index samples, colonies of *Enterococcus* were randomly sampled until 20 *E. faecium* clones per sample were isolated, requiring in some patients, isolation of up to 80 *Enterococcus* sp. colonies in order to obtain 20 that were *E. faecium*. The majority of patient samples (17/22) were not homogeneous, with 14/22 containing multiple enterococcal species and 9/22 containing a combination of vancomycin-resistant and vancomycin-sensitive *E. faecium* (Fig 2). In 5 patients we identified only VR *E. faecium* (Fig 2). For 1 patient (Patient 54) in the daptomycin group, only 8 colonies where isolated due to low *Enterococcus* densities in the patient sample.

## Exposure and resistance

Daptomycin exposure in the 6 months prior to the index sample ranged from 3 to 34 doses, and the most recent dose before the index sample was between 0 and 141 days earlier (see S1 Table). Patients with prior daptomycin treatment had greater proportions of resistant

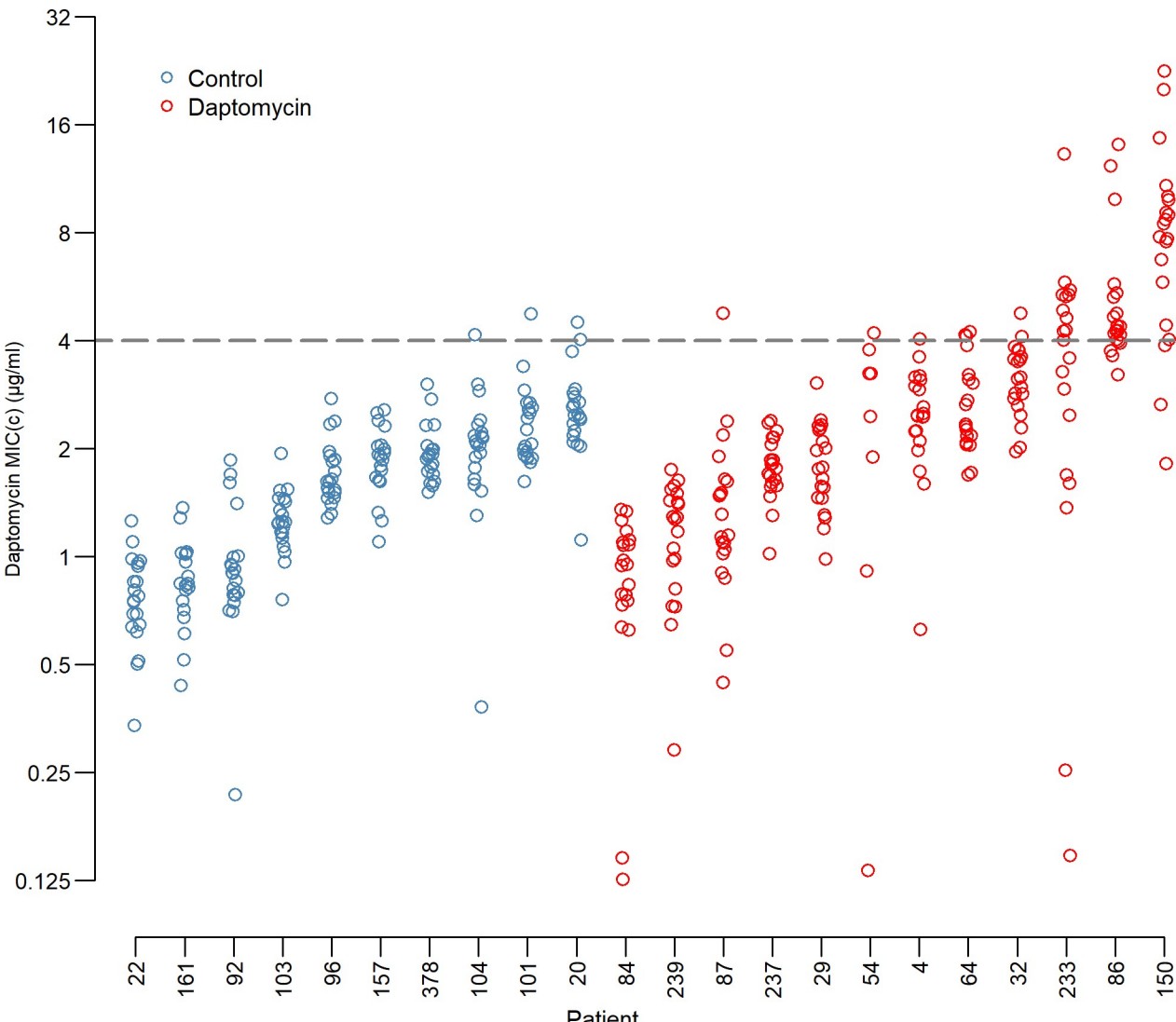

**Fig 3. Daptomycin MICs by patient.** Daptomycin resistance of each clone by patient and exposure group. Blue circles are the control patients and red circles are from daptomycin-exposed patients. Each circle is the mean of 2 independent estimates of the MIC$_C$ for a single clone. Clones above the dashed line are daptomycin resistant (due to the continuous nature of the computed MIC (MIC$_C$), an MIC$_C$ of >4μg/ml is equivalent to the clinical 2-fold MIC cutoff of ≥8μg/ml (see Methods for a more detailed discussion of the MIC$_C$ method)). The underlying data for this figure can be found in S1 Data.

clones, defined as a computed minimum inhibitory concentration (MIC$_c$—see Methods for a detailed description of this metric) greater than 4 μg/ml (Fig 3). Overall, 50 out of 426 clones were resistant to daptomycin with 94% of these in patients with prior daptomycin exposure. Eight of the 11 patients with resistant clones were from the daptomycin exposed group. Moreover, highly resistant *E. faecium* clones (MIC$_C$ >8 μg/ml) were only found in the daptomycin-exposed group. Patient mean MIC$_C$ was not associated with enterococcal species diversity (Spearman ρ = −0.06, *p* = 0.79; Fig A in S1 Text), the number of days since the last dose of daptomycin (Spearman ρ = −0.07, *p* = 0.83; S1 Fig), or the number of doses of daptomycin in the previous 6 months (Spearman ρ = 0.22, *p* = 0.50; S1 Fig).

We are also interested in how variation in the resistance phenotype is spread across the study population. A Bayesian mixed-effect model was designed to test for the presence of

**Table 1. Summary of Models and DIC analysis results.**

| Model | Mean deviance | DIC (pD) | ΔDIC | Patient effect | Distribution of clone effect different for each patient | Distribution of clone effect same for each patient | Distribution of clone effect depends on treatment group |
|---|---|---|---|---|---|---|---|
| Model 1: | 1,157 | 1,384 | 0 | Y | Y | | |
| Model 2: | 1,197 | 1,513 | 129 | Y | | Y | |
| Model 3: | 1,886 | 1,909 | 525 | Y | | | |
| Model 4: | 1,175 | 1,478 | 94 | | Y | | |
| Model 5: | 1,196 | 1,576 | 192 | | | Y | |
| Model A: | 1,201 | 1,468 | 84 | Y | | | Y |

Models 1 through 5 were compared using DIC criteria to ascertain how variation in resistance is spread across the study population. Models varied in the presence or absence of a patient effect and distribution of the clone effect. Model A was designed to test if clonal variation is affected by prior daptomycin exposure.

variation in daptomycin resistance at the levels of interest: within patients, between patients, and between groups (daptomycin exposed versus not daptomycin exposed). Using the Deviance Information Criterion (DIC) [24], the best fit model included a random effect for "patient," and for "clone" nested within patient, such that the distribution for the clone effect differed from one patient to another (Table 1).

The fixed effect for daptomycin exposure, $M_D$, provided support for the hypothesis that resistance is higher in daptomycin-treated patients with 94.5% of the posterior distribution for the fixed effect falling above zero (Fig 4 insert). The mean of $M_D$ is 0.57 on a $\log_2$ scale, which equates to approximately a 50% increase in mean $MIC_C$.

## Within-patient variation

The variation in $MIC_C$ among clones within patients differed from one patient to another (Table 1; Model 1 is a significantly better fit (as measured by a lower DIC) than Model 2). The

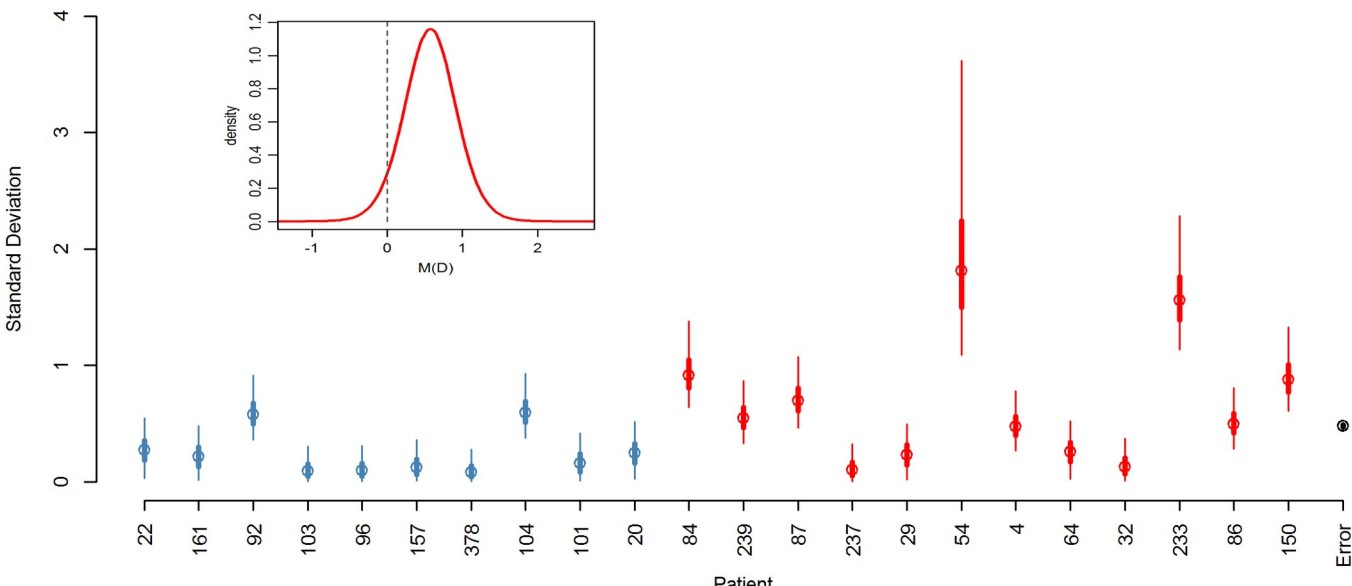

**Fig 4. Posterior distribution of random and fixed effects from the Bayesian mixed effects model.** (Main panel) Posterior distributions for standard deviations of the clones within patients and the error. Blue markers indicate control patients, red indicate daptomycin-treated patients, and black is the error (circle: mean; thick lines: 50% credibility interval; thin lines: 95% credibility interval). The 95% credibility interval of the error falls within the error marker. (Insert) Posterior distribution of fixed effect for prior daptomycin exposure. The underlying data for this figure can be found in S1 Data.

posterior distributions for the standard deviations of the clone effects for each patient are summarized in Fig 4. Further, Model A was designed to assess if there was evidence that patients with prior daptomycin exposure had greater between-clone variability in $MIC_C$. In particular, in Model A, the standard deviation for the "clone" effect for control patients was modeled with a single parameter $\lambda$ and for the daptomycin patients was $\lambda exp(\alpha)$ (see S2 Text for details). By using this parameterization, we were able to quantify the amount of evidence for daptomycin-treated patients having greater within-patient variability (by determining the proportion of $\alpha$'s posterior distribution that lies above zero). There is strong evidence for daptomycin-treated patients having greater within-patient variation than control patients with the estimated posterior distribution for $\alpha$ lying essentially above zero (mean = 1.04, 99% credibility interval is 0.677 to 1.798).

## Time series

Eight daptomycin patients and 6 control patients had more than 1 swab sample collected in 2016. While these additional swab samples do not allow for a comprehensive analysis of within-patient changes in resistance over time, they make possible a number of interesting case studies of within-patient changes in resistance. Ten *E. faecium* clones from each of these additional samples were isolated and tested for resistance. For standardization, the first 10 clones from the index samples in the previous analysis were used.

In all 3 patients where a sample was collected before the commencement of daptomycin treatment, the posttreatment sample contains clones that are more resistant than any clones sampled prior to treatment. The $MIC_C$ for the most resistant clone in the sample immediately prior to and posttreatment are: Patient 4 prior = 2.2 μg/ml, post = 4.2 μg/ml; Patient 87 prior = 1.7 μg/ml, post = 2.7 μg/ml; and Patient 150 prior = 0.5 μg/ml, post = 20.0 μg/ml (Fig 5). Patients 4 and 150 also showed an increase in mean $MIC_C$ (Patient 4: prior sample mean = 1.69 μg/ml, index sample mean = 2.86 μg/ml, t = −3.46, df = 18, *p* = 0.003; Patient 150: prior sample mean = 0.32 μg/ml, index sample mean = 13.08 μg/ml, t = −24.99, df = 18, *p* < 0.001). Patient 87 had a higher mean in the index sample but this was not statistically different to the pretreatment sample mean (prior sample mean = 1.15 μg/ml, index sample mean = 1.42 μg/ml, t = −1.41, df = 18, *p* = 0.18).

The maintenance of resistance following cessation of daptomycin treatment is highly variable between patients. In patients where a second sample was collected after the index sample (8 out of 12 patients), we see examples of long-term maintenance of resistance in the absence of further daptomycin exposure in our hospital (Patient 86) and rapid loss of resistance (Patients 4 and 150), which in both examples here is coupled with the presence of vancomycin-sensitive *E. faecium* which was not isolated in the index samples. In 2 patients, there was a slight increase in daptomycin resistance in the sample following the index sample despite no further daptomycin exposure, which was combined with a switch from predominantly vancomycin sensitivity to predominantly vancomycin resistance (Patients 64 and 233, see S2C and S2D Fig). In patients where the index sample contained no daptomycin resistant isolates, little change was observed after treatment ended (Patients 29 and 84, see S2C Fig).

## Genome analysis

To gain clearer insight to resistance evolution within patients, 4 patients were chosen that demonstrate the range of temporal pattern of resistance evolution described above. A total of 95 *E. faecium* isolates from 4 patients were sequenced (at least 5 clones per time point per patient—see Methods for more details). The core genome among these isolates includes 1,804 genes present in 100% of isolates, encoded in 1.68 Mb of sequence. An additional 2,795

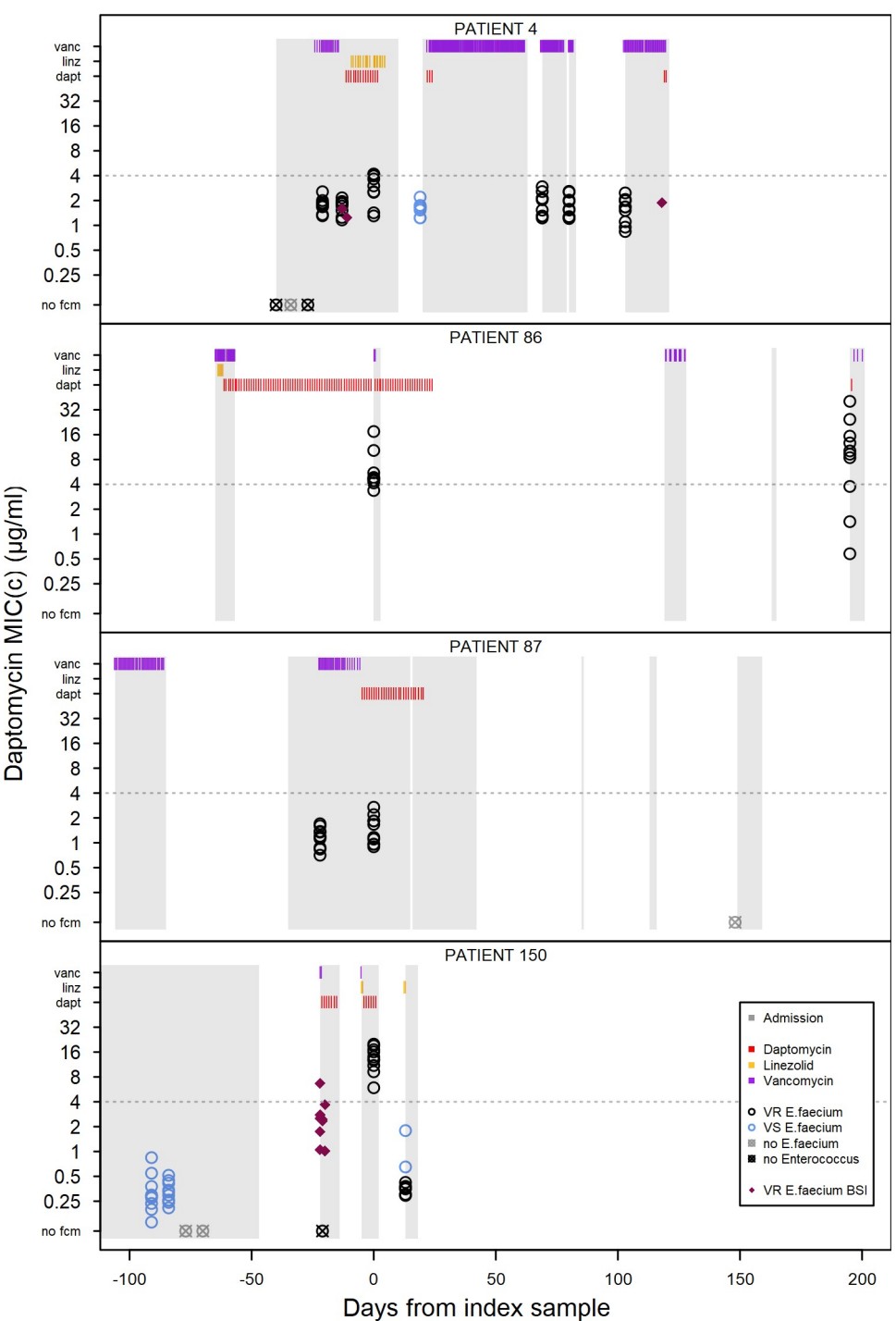

**Fig 5. Within-patient resistance over time.** Each plot shows patient admission periods, drug doses, and resistance of *E. faecium* clones isolated from screening swabs and blood stream infections for an individual patient. Time 0 is the index sample included in the previous analysis. Patient admission periods are shown as gray blocks, and individual doses of vancomycin (purple), linezolid (yellow), and daptomycin (red) are detailed in the bars at the top of the plot. Circles show daptomycin resistance (MIC$_C$) for 10 clones per sample. Each circle is the mean of 2 replicates with black circles denoting VR *E. faecium* and blue circles denoting VS *E. faecium*. Pink diamonds are isolates from VR *E. faecium* blood stream infections. The underlying data for this figure can be found in S1 Data.

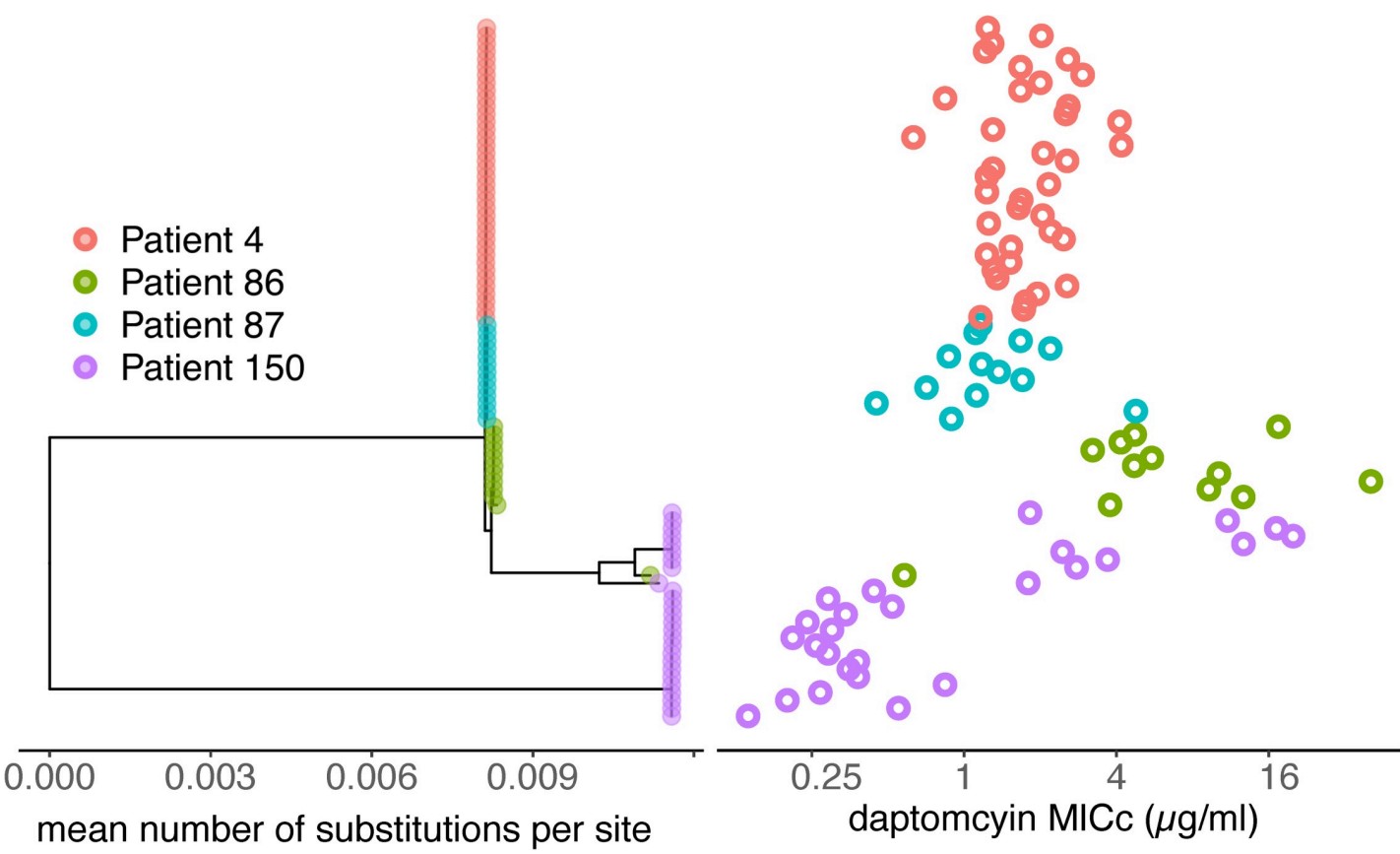

**Fig 6. Phylogeny for *E. faecium* strains isolated from 4 patients.** Midpoint rooted tree of 89 isolates. Six sequenced isolates were excluded that had core genomes identical to other samples in the alignment. The panel on the right shows the mean daptomycin $MIC_C$ (ug/mL) for each sample. Many isolates were closely related, resulting in very short branch lengths. See Fig 7 for more detailed visualization of phylogenetic structure within hosts. Sequence data is available at NCBI GenBank under the study accession number PRJNA673360.

accessory genes were annotated. This core genome was used to build a phylogeny (Fig 6). All genomes sequenced for this study fall within the previously defined hospital-associated *E. faecium* clade A [25].

*E. faecium* isolates were mostly clustered within patients, with 2 patients containing more than 1 cluster of isolates. Patient 150 contained 2 clusters separated by 20,062 SNPs, and Patient 86 contained a single isolate that differed from all the others from that patient by 2,997 SNPs, consistent with at least 2 introductions into these 2 patients. By contrast, the mean number of core genome SNPs between 2 isolates within the remaining clusters was 4, 30, 2, 6, and 14 SNPs for patients 4, 86, 87, Patient 150 cluster 1, and Patient 150 cluster 2, respectively, plausibly consistent with within-patient evolution. This study design cannot rule out many more introductions from a closely related donor, as the specific donor populations are unknown.

Phylogenetic patterns are largely consistent with the diversity of phenotypic trajectories in these 4 patients. In Patient 4, resistant isolates were only observed in the index swab, while later isolates were more sensitive. Concordantly, the resistant isolates from the index swab from Patient 4 were part of a single sub-clade which carried an N13S mutation in cardiolipin synthase A (ClsA), a mutation previously associated with daptomycin resistance [26] (Fig 7A; see Fig A and Table A in S3 Text for more details). The lack of resistance at later points in time reflects the lack of this sub-clade among the later samples. In Patient 86, there was long-lasting

resistance and diversity, and concordantly, we see that mutations in ClsA presenting in the index swab persisted in 4 of 6 isolates that were seen to be resistant later (Fig 7B; see Fig B and Table B in S3 Text for more details). Patient 150 was initially colonized with daptomycin- and vancomycin-sensitive *E. faecium*, then later by daptomycin-sensitive but vancomycin-resistant isolates (Fig 7C and S3 Fig). Sequencing revealed that the blood stream infection was caused by isolates from a second clade that subsequently gained daptomycin resistance in the gastro-intestinal isolates. The later reversion to a more daptomycin-susceptible population reflects the reemergence of the initial sequencing type, a subset of which are now vancomycin resistant and carry the vanA containing plasmid of the other clade (S3 Fig; see Fig D and Table D in S3 Text for more details).

Sequencing of Patient 87 isolates revealed more complicated dynamics than suggested from the phenotypic data (Fig 7D). Just 1 isolate from Patient 87's index swab had an $MIC_C$ above 4 µg/ml; however, all isolates sequenced from the index swab carried mutations in ClsA (see Fig C and Table C in S3 Text for more details). The resistant isolate and 2 others carried a ClsA R267H mutation previously associated with resistance, but in addition, the 2 sensitive isolates also carried a second mutation in ClsA, suggesting possible loss or reversion. Similarly, the second clade had a ClsA R221Q mutation; however, all isolates were sensitive, raising the question of whether the effects were modified by subsequent mutations.

The genetic analysis of these 4 patients is consistent with dynamic gain and loss of resistance in the off-target population and gives genetic explanations for even subtle resistance changes. However, mutations in known resistance genes do not explain all of the gains and losses of daptomycin resistance, and the sequencing suggests other candidates for further investigation (see Table A–D in S3 Text for a full list of mutations). Finally, the sequencing reveals dynamic gain and loss of vancomycin resistance. In Patient 4, the loss of vancomycin resistance was associated with loss of the entire plasmid containing the vanA cassette, or deletion of all or part of the cassette. In Patient 150, the gain of vancomycin resistance occurred by introduction of a second strain carrying the vanA cassette, or a transfer of the vanA containing plasmid from one strain to another.

## Discussion

Patients exposed to daptomycin have more daptomycin-resistant *E. faecium* in their intestines than unexposed patients. This difference is shown by the cross-sectional analysis (Figs 3 and 4 insert) and the patients with samples before and after daptomycin exposure (Fig 5). The findings support our hypothesis that intravenous daptomycin treatment leads to off-target selection for daptomycin resistance in the intestinal tract. Further, this finding is consistent with the ecology of this organism, with *E. faecium* most commonly found in the gastrointestinal tract and specifically adapted for transmission between patients in healthcare settings [27,28]. Transmission is by the fecal-oral route, which makes drug exposure and selection in the intestinal tract relevant to the risk of transmission of drug-resistant pathogens [29]. Daptomycin concentrations in the gastrointestinal tract are likely in the range that would select for resistance (see below), though this exposure is evidently not enough to eradicate the bacteria. When the results of this study are added to the ecology of *E. faecium*, it is plausible that off-target selection is a major mode of daptomycin resistance evolution in hospital.

Our study also sheds light on how *E. faecium* evolves in patients exposed to daptomycin. In the absence of daptomycin exposure, few patients had detectable variation in daptomycin resistance (Fig 3). With as little as 3 doses of daptomycin, the variation both within and across patients increased (Fig 4). Thus, there is either low-level preexisting variation in resistance that daptomycin exposure brings to the fore, or the supply of mutations is high enough during

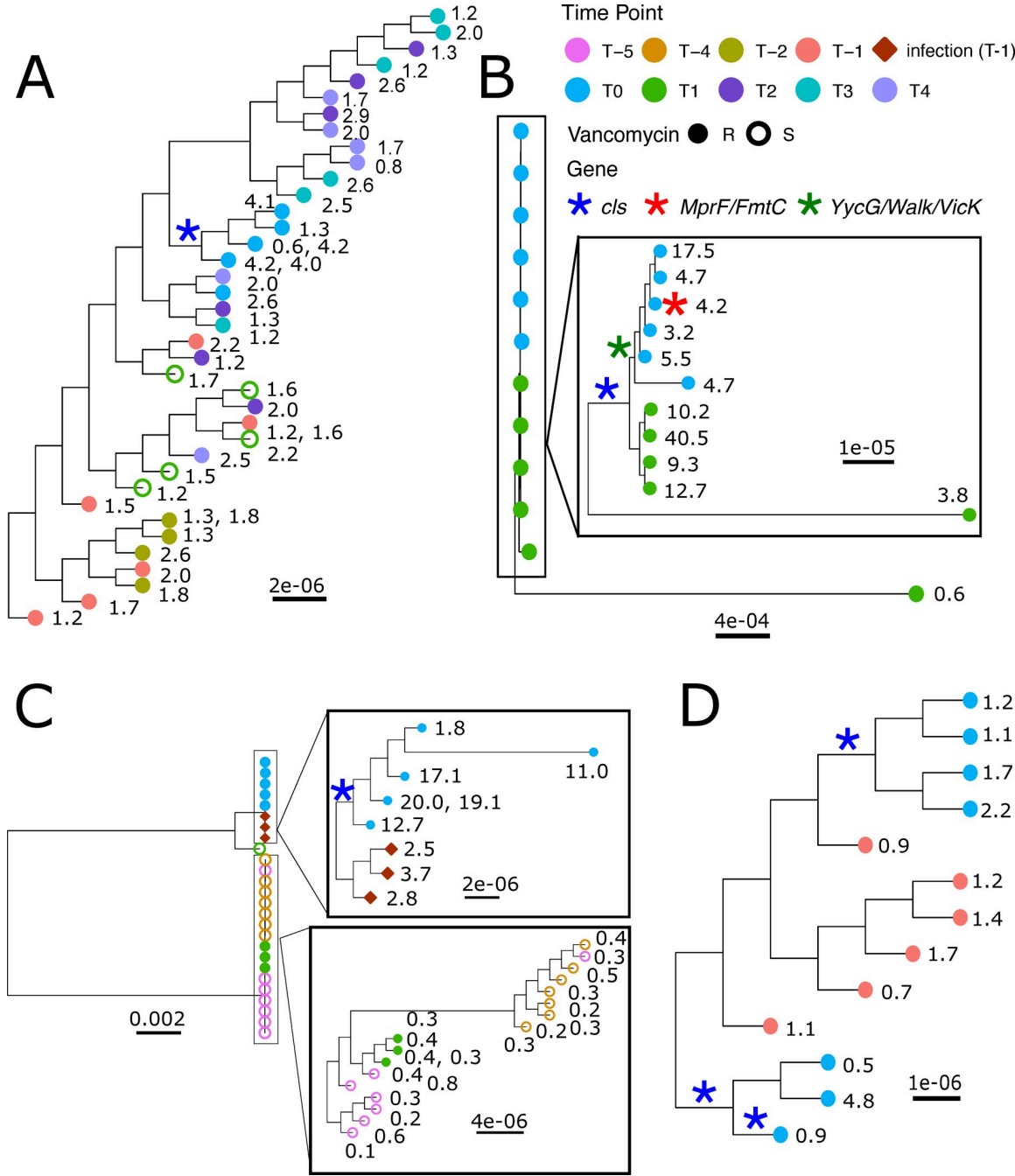

**Fig 7. Individual phylogenies for *E. faecium* strains isolated from 4 patients.** A midpoint-rooted phylogeny was created based on the core genome alignment of all isolates using RAxML. Here, subsets of the full tree (Fig 6) are shown, including only the isolates from each patient (i.e., other tips are masked). Scale bars for branch length (mean number of substitutions per site) are included for each tree and insert. Rectangle inserts show detailed structure of indicated groups. Daptomycin MIC$_C$s are listed for each tip. For tips representing 2 isolates with identical core genomes, the daptomycin MIC$_C$s for both isolates are listed. Asterisks indicate mutations in genes known to be associated with variation in daptomycin susceptibility. For each patient, one assembled genome (two for the 2 clades in Patient 150) was used as a reference genome for variant calling (see Methods). (A) Patient 4 (B) Patient 86 (C) Patient 150 (D) Patient 87. Sequence data is available at NCBI GenBank under the study accession number PRJNA673360.

treatment to result in a response to selection. In the 4 cases in which we obtained whole genome sequencing data, the results were more consistent with the latter. But, in either case, the response to selection is not limited by supply of resistant variants. Therefore, strategies such as mutation-prevention dosing which have the explicit goal of preventing the emergence of resistant variants may not work in the intestinal tract [30–34]. Alternative strategies that leverage competition to limit the spread of resistance may offer a way forward but often utilize lower doses [35–37], which raises the potential for conflict between controlling resistance in the gut and controlling resistance at the site of infection.

The variation in resistance within patients was greater after daptomycin exposure (Fig 4). In fact, all patients with resistant isolates continued to also contain clones below the resistance threshold. Thus, the raw material required for the population to return to a more susceptible state remained in the gut. Determining whether this variation in resistance is associated with variation in competitive ability in the intestinal tract, or transmissibility, may help identify better resistance management strategies. However, managing resistance in the intestinal population is fundamentally different from managing it in the target population. Orders of magnitude more bacteria may be present in the intestines [38], leading to a larger pool of variation as well as increased competition, both within species, and with other members of the microbiome.

Few data are available on the pharmacokinetics of daptomycin in the intestinal tract. In healthy adult men, about 5% of daptomycin is excreted in the stool over 72 hours [39]. Assuming stool water content of 100 to 200 ml/day [40] and 500 mg daily doses (6 mg/kg in an average US adult) [41], the daptomycin concentration in the stool could be 125 to 250 ug/ml, well above the MIC for most *Enterococcus*, though the active concentration may be significantly lower. It is unknown whether these calculations represent the drug level experienced by *Enterococcus* in the patient population in which VRE are often found, in whom liver and kidney dysfunction are common [42,43].

Temporal dynamics within patients were highly variable, even among the rather few cases with repeat sampling. In the 3 patients where we had a pretreatment comparison, resistance increased following treatment. Resistance was maintained in some patients even in the absence of continued drug treatment (Patient 86, Fig 5), while in other patients, populations reverted to being entirely sensitive (Patient 4: 17 days and Patient 150: 12 days; Fig 5). Thus, there are likely important unmeasured factors that influence the resistance dynamics in these patients, such as selection by other drugs. Likewise, it would be very interesting to know why daptomycin-resistant clones were not detected in all daptomycin-exposed patients (Fig 3), including the potential role of antibiotic tolerance as opposed to resistance in this setting. Finally, we note the limitations of a retrospective cohort study design including the possibility that selection could be due to collateral resistance or sensitivity.

Daptomycin resistance in *E. faecium* is a growing problem that has persisted despite efforts to minimize unnecessary drug use and prevent hospital transmission [23]. This study demonstrates that off-target evolutionary dynamics likely play an important role in this problem. In systems like this, where target and nontarget populations are compartmentalized, it will be challenging to identify dosing strategies that can optimally slow resistance emergence in both sites. Novel interventions that can separate treatment of infection sites from selection in off-target sites are more promising. These include interventions which neutralize the action of the drug in the gastrointestinal tract [6]. Given that many of the most serious antimicrobial resistance problems are caused by opportunistic pathogens, intervening in the evolutionary dynamics driven by off-target antimicrobial exposure in highly transmissible gastrointestinal carriage populations could have an outsized impact on the emergence and evolution of hospital-acquired resistant infections.

## Methods

### Study participants

A cohort study was conducted utilizing perirectal swabs from an infection prevention screening program at Michigan Medicine to determine the impact of intravenous daptomycin treatment on daptomycin resistance in gut populations of *Enterococcus faecium*. The study was approved by the University of Michigan Institutional Review Board. The initial inclusion criteria (see Fig 1) were all patients with a VRE positive (*E. faecium* or *E. faecalis*) swab using VRE Select agar (BioRad, Marnes-la-Croquette, France) in 2016 ($n = 618$). Patients included in the daptomycin exposure group had at least 3 administered doses of daptomycin (usually equal to 3 days of therapy) in the 6 months prior to the VRE positive swab (14 patients). Three doses was selected as a minimum requirement because treatment of enterococcal infections generally requires at least 3 days of treatment. The control group consisted of patients who had no known daptomycin exposure and at least 6 doses of linezolid (usually equal to 3 days of therapy due to twice daily dosing) in the last 6 months prior to a VRE positive swab (15 patients). For each patient, the first sample to meet the inclusion criteria was defined as the index sample. For each index sample, *Enterococcus* sp. clones were isolated until there were 20 *E. faecium* clones per sample. Samples where no *E. faecium* was isolated after sampling 20 random *Enterococcus* colonies, only 1 colony was isolated after sampling 40 random *Enterococcus* colonies, or where no *Enterococcus* was isolated were excluded from further analysis (Fig 1). The final data set included 12 patients in the daptomycin exposure group and 10 patients in the control group. For a further time-series analysis of these patients, up to 5 prior samples and all subsequent samples available from these patients in 2016 were tested for the presence of *E. faecium*. Ten random clones per sample were isolated from each of the *E. faecium*-positive samples. Finally, we collected isolates from all blood stream infection (BSI) in these patients.

### Ethics statement

This study was approved by the University of Michigan Institutional Review Board (ID no. HUM00102282), which determined that informed consent was not required as all samples utilized were collected for patient treatment purposes.

### Strain isolation

Perirectal swabs were obtained using E-swabs (BD) as part of the hospital VRE surveillance program. The swabs were first tested in clinical microbiology lab by streaking on VRESelect agar (BioRad) per manufacturer's recommendations. The swab was discarded, and the residual media was stored with glycerol (final concentration 20% v/v) at −80˚C. To isolate *E. faecium*, samples were streaked from the sample stored in glycerol onto Enterococcosel agar (BD BBL) in duplicate and incubated up to 72 h at 37˚C. The first 10 colonies from each plate (20 colonies per sample) were re-streaked on Enterococcosel agar and incubated for 48 to 72 h at 37˚C. One colony from each plate was then streaked on BHI agar (BD BBL) and a vancomycin (30 µg/ml Oxoid) disc was placed on each plate to determine vancomycin resistance. Plates were incubated for 24 h at 37˚C. One clone per plate was stored in BHI +20% glycerol at −80˚C.

To confirm the species of *Enterococcus*, a species-specific multiplex PCR was performed using primers for *E. faecium*, *E. faecalis* and VanA, VanB, VanC1, and Van C2/3 [44]. Briefly, 11.25 µl PCR Master Mix (iProof HF, BioRad), 50 uM of each primer, and 6.45 µl water (total volume 22.5 µl) per sample were combined. Sample was added as either 2.5 µl of bacteria in BHI glycerol taken from tubes prior to freezing, or 1 colony from a streaked culture of stored

bacteria. PCR was performed under the following conditions: 95˚C for 4 min; 30 cycles (98˚C for 10 s, 55˚C for 30 s, 72˚C for 30 s); 72˚C for 7 min. Gels were run for 1 h at 80 to 100 V on 2% Agarose in TAE buffer with 0.1 μl/ml SybrSafe.

Isolation steps were repeated until 20 *E. faecium* clones per sample or 10 clones for time-series only samples were isolated. Samples were excluded if more than 40 *Enterococcus* clones were isolated without any *E. faecium* (20 clones for time-series only samples), or no *Enterococcus* was detected after streaking the sample twice and then plating 80 μl of the initial patient sample on Enterococcosel agar (combined approximately 10% of the total sample volume). This sampling method resulted in a data set on species diversity within patients (Fig 2).

Ten patients had *Enterococcus* blood stream infections (BSI) within 6 months of the index swab sample, and a total of 45 isolates were taken from these patients. Blood samples were cultured in blood bottles and streaked on Chocolate agar in the clinical microbiology lab. Single colonies were streaked on BHI agar 3 times, and then a single colony was stored in BHI +20% glycerol.

## MIC testing

MIC testing was performed by broth microdilution (BMD) according to CLSI M7 guidelines [45]; each sample was tested in duplicate and 1 of 4 patient-derived *E. faecium* strains was included on each run as a positive control. All clones were initially tested on plates containing 2-fold dilutions of daptomycin with final concentrations ranging from 0.125 μg/ml to 16 μg/ml, and the optical density (OD) of each well (600 nm) was determined by plate reader (FLUOstar Omega, BMG Labtech, Ortenberg, Germany). A Hill function was fitted to the OD values for each dilution series, and resistance was measured as the concentration at which this curve crossed a defined cutoff. We set the cutoff at 2 standard deviations above the mean of the negative control wells (see S2 Text). This threshold was chosen because it reflects the automated equivalent of a visual MIC, i.e., the point at which the OD falls below the detectable limit. Thus, we refer to this value as the computed MIC ($MIC_C$) as it is the minimum antibiotic concentration required to reduce the OD reading to the background level and is reported as a continuous (not 2-fold) value. If the initial concentration range did not contain at least 2 concentrations above and below the cutoff, the assay was repeated on either increased (1 μg/ml to 64 μg/ml) or decreased (0.0625 μg/ml to 4 μg/ml) concentrations as appropriate. Individual assays of clones were also excluded if the Hill curve did not fit the data points well, determined as an $MIC_C$ greater than one 2-fold dilution from the lowest concentration with an OD below the cutoff.

## Statistical analysis of *$MIC_C$*

We analyzed the $log_2$ of the $MIC_C$ values using Bayesian mixed effect models [46]. Models included a fixed treatment effect for patients that fulfilled the daptomycin exposure case definition (see above). To quantify the evidence for prior daptomycin exposure increasing $MIC_C$, we determined the proportion of the posterior for the fixed "treatment" effect that was above zero. The full model included 23 random effects (1 "patient" and 22 "clone" effects) and allowed the distribution of the "clone" effect to depend on patient. We fitted 6 candidate models which considered different combinations of the random effects. In addition to testing different combinations of the random effects from the full model, we also considered models where the distribution of the "clone" effect was identical for all patients, and a model where this distribution depended on treatment group (see Table 1 and S2 Text for details). All random effects are assumed to be normally distributed with zero mean and standard deviations estimated using the MCMC program JAGS [47,48]. Uninformative priors were used (see Table B in S2 Text).

We ran each model for $20 \times 10^6$ iterations with a burn in of $10 \times 10^6$ steps and a thinning interval of $2 \times 10^3$. This resulted in $5 \times 10^3$ parameter samples for each model run. This process was repeated to generate 4 chains, with randomly chosen initial starting values, for each model. Posterior convergence was confirmed in 2 ways: (1) empirical inspection of the estimated posterior distributions (all 4 chains resulted in very similar distributions); and (2) the Gelman–Rubin convergence diagnostic (this statistic was essentially 1 for all parameters, which is consistent with the chains having converged).

The models were then compared using the Deviance Information Criterion (DIC) [24]. The relative fits of the models are summarized using ΔDIC scores which are the differences in DIC between the best model and each alternative model. Although there is no universally agreed threshold for significance of ΔDIC scores, there is precedent for treating ΔDIC scores greater than 10 as providing very little support for the model [49] (similar to rules used for the Akaike Information Criterion (AIC) [50,51]). The smallest ΔDIC score was 84, indicating that Model 1 is clearly the preferred model. To assess how well the data fit the best model (as selected by DIC comparisons), we used the posterior distributions of the selected model to generate synthetic data sets and examined the distributions of a number of different summary statistics (see S2 Text).

## Genome sequencing

**Bacterial isolates.**   A total of 95 *E. faecium* isolates were sequenced from the 4 patients shown in Fig 5. Patients were sampled at multiple time points. For each perirectal swab sample, the first 5 random *E. faecium* isolates were sequenced. In addition, the isolates with the highest and lowest daptomycin $MIC_C$ were included from each sample. For each sample, the first isolate that was vanR, vanS, or van(discordant) was also included. For Patient 150, 1 blood isolate from the first order of each day was also included. Isolates sequenced from each patient were as follows: 42 isolates from Patient 4, 12 isolates from Patient 86, 13 isolates from Patient 87, and 28 isolates from Patient 150.

**DNA sequencing.**   Whole genomic DNA preparations were submitted to the University of Michigan sequencing core for Illumina library preparation and paired end 150 bp Illumina NovaSeq. A subset of samples was additionally sequenced using the Oxford Nanopore Minion. Nanopore libraries were prepared using the Nanopore Ligation Sequencing Kit (SQK-RBK-004) and sequenced using R9.4.1 flow cells with Guppy 3.2.10 using fast base calling mode.

**Genome assembly.**   Quality control of sequencing reads was performed using Trimmomatic [52]. Reads were assembled using Unicycler [53] and annotated with Prokka [54]. The core genome was analyzed with Roary [55]. Additional hybrid assemblies, generated using long and short read input data, were created for reference genomes used in variant calling (details below).

**Phylogenetic analysis.**   Nucleotide substitution models were evaluated for best fit to our data using jModelTest2 [56,57]. A tree was constructed with RAxML [58] using a generalized time reversible model with a proportion of invariable sites and variable rates across sites (GTR + I + G) [59–61]. The input was the core gene alignment generated with Roary, excluding 6 samples with core genomes exactly identical to other sequences in the alignment. The tree was midpoint rooted and visualized using ggtree [62]. The PopGenome [63] package was used to determine pairwise nucleotide diversity within and between populations of isolates.

**Variant calling.**   To identify genomic variants, trimmed reads from each sample were mapped against an assembled *E. faecium* genome originating from the same patient. For comparison of isolates from Patient 4, Patient 86, Patient 87, reference genomes were from isolates PR01996-12, PR00859-7, and PR02395-7, respectively. Two isolates were used as references for 2 clades within Patient 150: PR05720-3 and PR02648-8. Reference genomes were assembled

using both long and short read data from isolates. For variant calling, short reads were aligned to references using Burrows-Wheeler Aligner (BWA) [64], and candidate variants were identified with The Genome Analysis Toolkit (GATK) [65]. Reference contigs <500 bp in length were excluded from variant calling analysis. This resulted in 3.18 Mb analyzed for Patient 4, 3.33 Mb analyzed for Patient 86, 3.29 Mb for Patient 87, and 3.12 Mb (PR05720-3) or 2.90 Mb (PR02648-8) for Patient 150. Reads from the reference sample were aligned to the reference genome (aligned to self) to generate a list of background variants; these background variants were filtered out during variant calling. Additionally, candidate variants were filtered out if coverage was <10, or if the alternate allele was called in <70% of reads. Remaining candidate variants were screened by visual inspection of alignments in Tablet [66]. Variants were manually assigned to branches on phylogenies based on parsimony with reference to an outgroup sample. To screen for larger deletions, BEDTools [67] was used to identify regions >100 bp with zero coverage. To investigate the gain of vancomycin resistance in several isolates from Patient 150, vancomycin-resistant isolates were mapped against a second reference from the same patient (PR05720-3) to compare the *vanA*-containing plasmids. As an additional check for variants in genes associated with daptomycin susceptibility, trimmed reads from each sample were also mapped against selected loci from the *E. faecium* reference genome DO that have previously been associated with daptomycin resistance [68]. Results from mapping against the DO reference were consistent with the original variant calls; no additional variants were identified in this check.

## Supporting information

**S1 Fig. Relationship between daptomycin exposure and resistance.** Mean daptomycin $MIC_C$ per patient by (left) the number of days since the last dose and (right) the total number of daptomycin doses in the 6 months prior to the index sample. For underlying data see S1 Data.
(TIF)

**S2 Fig. Time-series Plots for All Patients.** Time-series for each patient showing patient admission periods, drug doses, and resistance of *E. faecium* clones isolated from screening swabs and all *Enterococcus* blood stream infections. Patient admission periods are shown as gray blocks, and individual doses of vancomycin (purple), linezolid (yellow), and daptomycin (red) are detailed in the bars at the top of the plot. Circles show daptomycin resistance ($MIC_C$) for 10 clones per sample. Each circle is the mean of 2 replicates with black circles denoting VR *E. faecium* and blue circles denoting VS *E. faecium*. Diamonds are isolates from VR *E. faecium* (purple), VS *E. faecium* (blue), and VS *E. faecalis* (gray) blood stream infections. Panel A–Patients 20, 22, 92, and 96; Panel B–Patients 101, 103, 104, 157, 161, and 378; Panel C–Patients 4, 29, 32, 54, 64, and 84; and Panel D–Patients 86, 87, 150, 233, 237, and 239. For underlying data see S1 Data.
(PDF)

**S3 Fig. Schematic of within-patient dynamics for Patient 150.** Based on the phylogenies in Fig 7, isolates from Patient 150 have been assigned to 1 of 3 clades. Clades were detected in sampling when they are noted above a time point and not detected if there is a gray ⊗. While Clade 1 was not detected in samples between T-3 and T0, it appears to have persisted through to T1, where it has acquired a VanA plasmid from Clade 2. Clade 2 was first isolated from the blood stream; however it is likely that this came from an undetected gastrointestinal population.
(TIF)

**S1 Table. Numbers of doses of daptomycin in the 6 months prior to the index samples and the number of days between the most recent dose and the index sample.**
(PDF)

**S1 Text. Sample species diversity.** Distribution of *Enterococcus* species with patient samples and figures showing relationship between Shannon Diversity and mean or max MIC$_C$s and within-patient diversity.
(PDF)

**S2 Text. Supplementary methods and analysis.** Full details of methods used to derive MIC$_C$ values and full descriptions of models used for analysis.
(PDF)

**S3 Text. Genome analysis.** Full lists of nonsynonymous mutations identified within patients and their locations on the phylogenies from Fig 7.
(PDF)

**S1 Data. Data for reproducing all main and supporting figures.**
(XLSX)

## Acknowledgments

We thank Amit Pai for discussion about pharmacokinetics.

## Author Contributions

**Conceptualization:** Clare L. Kinnear, Andrew F. Read, Robert J. Woods.

**Data curation:** Clare L. Kinnear, Elsa Hansen, Kevin C. Tracy.

**Formal analysis:** Clare L. Kinnear, Elsa Hansen, Valerie J. Morley, Kevin C. Tracy.

**Funding acquisition:** Andrew F. Read, Robert J. Woods.

**Investigation:** Clare L. Kinnear, Meghan Forstchen.

**Methodology:** Clare L. Kinnear, Elsa Hansen, Andrew F. Read, Robert J. Woods.

**Project administration:** Clare L. Kinnear, Robert J. Woods.

**Resources:** Andrew F. Read, Robert J. Woods.

**Supervision:** Andrew F. Read, Robert J. Woods.

**Validation:** Clare L. Kinnear.

**Visualization:** Clare L. Kinnear, Elsa Hansen, Valerie J. Morley.

**Writing – original draft:** Clare L. Kinnear, Elsa Hansen.

**Writing – review & editing:** Clare L. Kinnear, Valerie J. Morley, Andrew F. Read, Robert J. Woods.

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
