## [Editor Report · Decision Letter 0]

5 Feb 2020

Dear Dr Kinnear, 

Thank you for submitting your manuscript entitled "Antimicrobial treatment impacts resistance in off-target populations of a nosocomial bacterial pathogen: a case-control study." for consideration as a Research Article by PLOS Biology.

Your manuscript has now been evaluated by the PLOS Biology editorial staff as well as by an academic editor with relevant expertise and I am writing to let you know that we would like to send your submission out for external peer review.

Please re-submit your manuscript within two working days, i.e. by Feb 07 2020 11:59PM.

Kind regards,

Lauren A Richardson, Ph.D

Senior Editor

PLOS Biology

---

## [Decision Letter · Decision Letter 1]

27 Feb 2020

Dear Dr Kinnear,

Thank you very much for submitting your manuscript "Antimicrobial treatment impacts resistance in off-target populations of a nosocomial bacterial pathogen: a case-control study." for consideration as a Research Article at PLOS Biology. Your manuscript has been evaluated by the PLOS Biology editors, an Academic Editor with relevant expertise, and by three independent reviewers.

The reviews of your manuscript are appended below. You will see that the reviewers find the work potentially interesting. However, based on their specific comments and following discussion with the academic editor, I regret that we cannot accept the current version of the manuscript for publication. We remain interested in your study and we would be willing to consider resubmission of a comprehensively revised version that thoroughly addresses all the reviewers' comments. We cannot make any decision about publication until we have seen the revised manuscript and your response to the reviewers' comments. Your revised manuscript would be sent for further evaluation by the reviewers.

You'll see that while both reviewers #1 and #2 are somewhat positive about the study, reviewer #3 is more negative, saying that the central findings are rather unsurprising. Both #1 and #3 explicitly think that there are gaps in the study, with both saying that for a significant study of this type one would nowadays expect genome sequences and further analyses. In addition, #2 questions the theoretical framing of the study. All three express concerns about the use of linezolid as a control, or feel that you failed to capitalise on the use of this cohort. Having discussed the reviews with the Academic Editor, we think you should address all of the concerns raised by the reviewers, with the analysis of isolate genome sequences as a prerequisite.

We appreciate that these requests represent a great deal of extra work, and we are willing to relax our standard revision time to allow you six months to revise your manuscript.

We expect to receive your revised manuscript within 6 months. Please email us (plosbiology@plos.org) if you have any questions or concerns, or would like to request an extension. At this stage, your manuscript remains formally under active consideration at our journal; please notify us by email if you do not intend to submit a revision so that we may end consideration of the manuscript at PLOS Biology.

**IMPORTANT - SUBMITTING YOUR REVISION**

*Resubmission Checklist*

*Published Peer Review*

*PLOS Data Policy*

*Blot and Gel Data Policy*

Sincerely,

Roli Roberts

Senior Editor

PLOS Biology

REVIEWERS' COMMENTS:

Reviewer #1:

The manuscript by Kinnear and colleagues examines how IV antibiotic use can drive selection of resistance in a bystander population of intestinal bacteria. The case control design and large numbers of clones tested from each patient are clear strengths of the study. Study limitations include a lack of connection between phenotypic and genotypic resistance, and ambiguity regarding some of the methods. The manuscript could be improved by addressing the following comments:

Major comments:

1. Phenotype-genotype connections. The authors report changes in daptomycin susceptibility phenotypes, but the lack of genotype data makes the paper seem very limited in scope. The authors went to great lengths to isolate numerous clones from their study patients, and selective genome sequencing would provide valuable insight into the population dynamics underlying the observed phenotypic changes. The patients described in lines 181-183 would be prime candidates for targeted whole-genome sequencing, and including this information would greatly increase the impact and relevance of this work.

2. Focus on daptomycin and not linezolid. The authors use patients treated with linezolid as a control group, but presumably the same bystander selection could be happening in the linezolid-treated patients as well. Why not include information about how linezolid susceptibility changes as well, and have the two drug treatment groups serve as controls for one another? Do the authors believe that the evolutionary dynamics are different with linezolid versus daptomycin?

3. Lack of clarity in methods. This happens in two places: First, the paragraph describing isolation of clones (lines 101-114) is confusing, especially because it follows immediately after mention of screening for VRE. A topic sentence stating that VRE-positive rectal swab samples were used to isolate for multiple VR E. faecium clones from each patient would be helpful. Expanding Figure 1 to include the screening/isolation of clones from each patient would also increase clarity. Second, use of the term "computed MIC" is likely to create confusion for readers accustomed to seeing MIC values follow a standard log2 scale. As described it sounds like the authors have calculated a value that more closely resembles an IC90, rather than an MIC. Unless there is literature precedent for the term "MICc" (which if so should be cited), using a different term to quantify drug susceptibility would help avoid confusion.

Minor comments:

1. The title is very broad and should probably be modified to include "Daptomycin" and "VRE."

2. Consider including the concept of antibiotic tolerance in the Discussion. If daptomycin concentrations in the GI tract indeed reach >100ug/mL, then VRE with greater tolerance (which may or may not also have greater resistance) will be the ones to survive. For this reason, assays that quantify drug tolerance/killing might be more appropriate that an assay looking for overt "resistance" in this setting.

3. The information on species diversity included in the S1 Appendix is worthy of inclusion in the main text, and its inclusion would increase the relevance of the study. Did species diversity change pre- vs. post-daptomycin exposure? Either result would be interesting, and should be discussed.

4. Why were at least six doses of linezolid required for inclusion in the control group? Versus only three or more doses of daptomycin.

Reviewer #2:

Kinnear et al. investigate the extent of daptomycin resistance in enterococcal isolates from individuals with daptomycin exposure within the past 6 months and those without recent daptomycin exposure. They report that dapto exposure is associated with increased dapto resistance, that treated patients have a greater amount of within host diversity in the extent of resistance, and that the duration of colonization with dapto resistant VRE varies. Strengths of the study include its direct assessment of resistance in carried isolates and its comparison across multiple individuals recently dapto exposed / unexposed. 

Although framed as an analysis of 'off-target' effects, the analysis is conceptually similar to defining the risk factors for resistance to a given antibiotic—and in that way, finding that exposure to daptomycin results in selection for colonization with daptomycin resistant organisms is in keeping with what has been observed for many other antibiotic-bacteria pairs. Similarly, the finding that duration of colonization with the resistant bacteria vary recapitulates what has been found in other analyses, including for VRE (https://www.ncbi.nlm.nih.gov/pubmed/12002235). It would be helpful if the authors would put their findings in the context of what's known about selection for other kinds of resistance in enterococcus and other GI-colonizing bacteria and how those isolates transmit. How do the authors' findings confirm, extend, and / or contrast with expectations based on the literature? 

The title refers to this as a case-control study. My epidemiological terms may well be rusty - could the authors clarify why this is a case-control, as opposed to a cohort study, given that the two groups were selected on the basis of different exposures?

A key part of the questions the authors set out to address has to do with resistance phenotype (as in the paragraph beginning line 69). The authors chose to generate a 'computed' MIC rather than stick with the standard MIC—could the authors please justify this choice? Given the expectation of error in MIC by broth microdilution testing for daptomycin in Enterococcus (Campeau et al., AAC 2018), how do the authors propose to allay concern that the analysis is overinterpreting the MICc and the variation they calculate, given the errors in measurement? Or do the authors think that the small (<~2 fold) MIC variations as plotted in Fig 2 are real and there are genetic/physiologic mechanisms that explain these variations (and if so, what support is there for this interpretation)? 

In that paragraph, the authors also articulate a foundational hypothesis for the study, but it's a bit confusing. Could the authors clarify what they mean by "most transmitted" daptomycin resistance is due to off-target selection? It seems like this manuscript doesn't investigate transmission, but instead focuses on within-host dynamics of daptomycin resistance. Also - what's the alternative hypothesis, and wouldn't many of the questions pertain even if the hypothesis is false (even a small amount of transmitted resistance comes from the gut)? 

Risk factors for VRE colonization include hospitalization and antibiotic use (as well demonstrated in Fig 4, where the patients have multiple hospitalizations over the course of months). How does use of other antibiotics (e.g., cephalosporins, a known risk factor for VRE) impact their observations? Could there be selection for the dapto resistant strains through co-resistance? (e.g., to quinolones or other antibitoics?) 

Were the patients in the Time Series section drawn from those in the rest of the study? If so, could you identify in Figure 4 which samples contributed to the other sections of the study? 

Why was 6 months the limit for evaluating prior exposure? If you look back further, were there other exposures in either cohort? 

Minor comments

Line 30. Suggest reporting "50% higher" in terms of MIC dilutions

Lines 70-72. I'd guess there are studies that look for risk factors for daptomycin-resistant VRE, including past daptomycin use (e.g., https://www.ncbi.nlm.nih.gov/pmc/articles/PMC6176497/). Are the authors suggesting that evaluation of resistance has only been identified in the population causing an infection, and that population then persists in some way after treatment for infection and seeds future infections? 

Line 110-3. Construction is confusing: how is 9/22 isolates with a combination of VRE and VSE a majority?

Methods - what if anything was done with the isolates from blood stream infection?

Suggest clarifying in Figure 1 the 3 days of therapy. Why was 3 days chosen as the lower limit?

Fig 2 legend. What do the authors mean by 'equivalent to the clinical 2-fold MIC cutoff'? I think the authors are trying to clarify that, given that MICs are only determined in 2 fold dilutions, an MIC of >4 will appear as ≥8 - is that correct? However, it seems they might be suggesting that an MIC of 4 is interpreted as an MIC of 8 in clinical settings.

Fig 4. What is "fcm"? Please explain all abbreviations in the Fig 4 legend. Patient 150 seems to have a bunch of VSE without any VRE at ~-100. How were these obtained, if inclusion in the study required VRE?

Fig 2 and Supplemental Fig S1b look identical. 

Fig S2-2. Please label the MICs, rather than just using relative size. 

Reviewer #3:

The submission by Kinnear et al. examines the important question of how antibiotic treatment using daptomycin on E. faecium may select for increased daptomycin resistance within the E. faecium population of the gut essentially laying the foundation for new daptomycin resistant strains as an unintended consequence. From a collection of rectal swabs within a single hospital site 6,726 patients were screened for vancomycin resistant enterococci (VRE). Of these, 618 were VRE positive. It is not clear why only VRE strains were of interest (though the clinical relevance is clear). Off target effects could have been more broadly survey initially from the broader enterococci and then the authors could drill down on the VRE strains to see if there was something special genetically (and thus phenotypically) that would be characteristic in pre-disposing their ability to become resistant to daptomycin. From this group 23 had been treated with daptomycin and made up the initial study group with another group treated with linezolid used as a control (n=15). Strains were isolated and the MIC to daptomycin determined using standard CLSI methodology. As expected the patient group that had been treated previously with daptomycin held a substantially more diverse population of daptomycin resistant phenotypes presumably as a result of off-site selection.

 Of interest is the fact that daptomycin is administered by IV and not orally so the compartmentalization of the drug might have suggested that off site deletion would be less. Clearly, orally administered antibiotics get into the gut and there have been studies showing off site selection in that more expected context. The authors cite a previous work that 5% of the administered daptomycin can be observed in the feces of patients but I found the estimate of 125 to 250 micrograms/ml extraordinarily high seeing as the clinical breakpoints are in single digits. Nearly all Gram positives would be eliminated by such extremely high doses. Ref. 37 is about a study where C14-daptomycin was administered to patients and recovery of C14 in urine and feces was tested. Fig. 2 in that paper states that cumulative C14 recovered from feces was 5% but it does not mean that the recovered C14 was intact daptomycin. In fact, the same figure shows that C14 recovery in urine was 78% and that corresponded to 57% daptomycin. 

 Unfortunately there are quite a few gaps in the study. Sequencing and phylogenetics of isolated strains are to be expected in this day and age and none is performed in this work. Many wonderful studies have been performed with single and systems of hospitals tracking the movement of strains such as A. baumannii from beds to workers etc. This paper merely shows that patient treated with daptomycin are potential reservoirs for reinfection or colonization of the hospital across patients/locations. This does not surprise the reviewer as it has been shown in many other cases. The author speculate (lines 69-75) that off target selection and the associated evolutionary dynamics would be important to know but they have the collection of strains and expertise to determine and test that hypothesis.

General comments on the paper:

1. What is the normal level of diversity in daptomycin resistance in the gut flora? The test group in this study included patients treated with daptomycin and the control group had patients treated with linezolid. The linezolid in the control group could have cleared some of the daptomycin resistant members of the gut in the control population. Another control group of individuals not treated with antibiotics would have been more informative. 

(The gut bacteria are exposed the host derived cationic antimicrobial peptides that are similar in mechanism to daptomycin. This exposure can also induce resistance and thus, a control patient population not treated with any antibiotics will be a better baseline for this study). 

2. It is surprising that E. faecium isolates from only 3 out of the 12 daptomycin treated patients showed high levels of daptomycin resistance. A very small number of clones from other patients crossed the MIC threshold of 4μg/ml and even so, they did not differ very much from the control patients 104, 101 and 20. 

3. Lines 202-211: Among the 12 patients tested, data was only available from 3 patient samples to determine if daptomycin resistance was maintained after cessation of treatment. Within this small sample size, clones from 2 of the 3 patients showed rapid loss of daptomycin resistance. Did the authors measure the fitness cost of acquiring daptomycin resistance? This could provide some insight into the reason for the rapid loss of resistance. 

4. In line 69, the authors hypothesize that off-target selection in the GI tract is the cause for most transmitted daptomycin resistance in VRE. However, this work shows no association between the off-target daptomycin resistance in fecal samples and the transmission of daptomycin resistance. Yet, in line 224 it is stated that this could be a plausible cause of evolution of daptomycin resistance in hospital. This statement seems a bit far-fetched, especially when resistance was rapidly lost in 2 out of 3 patients whose long-term samples were available.

---

## [Decision Letter · Decision Letter 2]

20 Oct 2020

Dear Dr Kinnear,

Thank you for submitting your revised Research Article entitled "Daptomycin treatment impacts resistance in off-target populations of vancomycin-resistant Enterococcus faecium, a nosocomial bacterial pathogen." for publication in PLOS Biology. I have now obtained advice from two of the original reviewers and have discussed their comments with the Academic Editor. 

Based on the reviews, we will probably accept this manuscript for publication, assuming that you will modify the manuscript to address the remaining points raised by the reviewers. Please also make sure to address the data and other policy-related requests noted at the end of this email.

IMPORTANT:

a) Please attend to my Data Policy requests below.

b) Please address the remaining requests from reviewers #1 and #2.

c) Please truncate the title to "Daptomycin treatment impacts resistance in off-target populations of vancomycin-resistant Enterococcus faecium"

We expect to receive your revised manuscript within two weeks. Your revisions should address the specific points made by each reviewer. In addition to the remaining revisions and before we will be able to formally accept your manuscript and consider it "in press", we also need to ensure that your article conforms to our guidelines. A member of our team will be in touch shortly with a set of requests. As we can't proceed until these requirements are met, your swift response will help prevent delays to publication.

- a cover letter that should detail your responses to any editorial requests, if applicable

*Copyediting*

*Published Peer Review History*

*Early Version*

Sincerely,

Roli Roberts

Senior Editor,

rroberts@plos.org,

PLOS Biology

DATA POLICY:

Please submit your sequence data to GenBank or equivalent. In addition, we ask that all individual quantitative observations that underlie the data summarized in the figures and results of your paper be made available in one of the following forms:

Regardless of the method selected, please ensure that you provide the individual numerical values that underlie the summary data displayed in the following figure panels as they are essential for readers to assess your analysis and to reproduce it: Figs 2, 3, 4, 5, 6, 7, S1-1, S1-2, S2-1, S2-2, S3 Figs, S4-1 through S4-7, S5-1 through S5-4. NOTE: the numerical data provided should include all replicates AND the way in which the plotted mean and errors were derived (it should not present only the mean/average values).

REVIEWERS' COMMENTS:

Reviewer #1:

The manuscript is much improved from the initial submission, and the inclusion of the genome sequence data greatly increase the importance of the work. I only have a few minor comments:

1. Consider abbreviating "computed MIC" as cMIC, rather than MICc. 

2. Line 71: "VRE" is not defined before this use. Consider changing to VR E. faecium.

3. Figure 6 - is the MIC axis title meant to be MICc/cMIC?

4. Figure 7 - are the blue asterisks primarily meant to show clsA mutations? I suggest adding a blue asterisk to the figure key, and also listing the actual mutations themselves on the phylogenies. Also in the text Fig. 7D is discussed before Fig. 7C - either the figure panels should be flipped or the text should be re-organized. Finally, a simple schematic of the bacterial population dynamics over time in Patient 150 as revealed by WGS would be helpful.

Reviewer #2:

The authors have substantially revised their manuscript, adding genome sequencing and phylogenetic data. They expand on their main point -- that use of daptomycin selects for increased resistance in gut-resident enterococci ("off target" effect), which are transmissible -- with comments on the dynamics and variability of phenotypic resistance and genotypic evidence in support of within-host evolution of resistance. 

I appreciate the authors addressing many of the items raised in the first round of review. For my point about resistance to other antibiotics, it wasn't that I expect daptomycin resistance to confer resistance to other antibiotics; instead, I meant co-resistance: isolates that develop dapto resistance may be selected for if they are also resistant to, say, quinolones, and quinolones are also used on the patients. To the extent that there is variation in susceptibility to other drugs and use of those drugs in patients, this may explain variations in persistence of daptomycin resistant lineages. 

On the genomics analysis, I have a few questions and requests for clarification.

- Fig 6. Why leave out 6 isolates? It seems many have identical core genomes based on this phylogeny. If there are sequence differences in these isolates that are hidden by the colors, then I'd ask for a different visualization that allows the reader to see genetic differences among the isolates. 

- Fig 7 (and S5 appendix). Please specify in the legend what it is we're looking at. This looks like a cladogram. Is it? What are the branch lengths? How was the tree generated? How is it rooted? 

- Fig 7 key. "Infection" is not a timepoint. If the idea was to give a sense of temporal progression, could you please present a time-based tree?

- The amount of within-host diversity seems substantial, given the expected mutation rate of ~7 snps/genome / year. 

- I'm confused as to the variant calling approach. Why were variants called against two references? If there were discrepancies, how was this dealt with? 

- Please make clear which references and methods were used for generating each of the trees. It seems Fig 6 was made form a core genome alignment, but Fig 7 and S5 may have been from reference-based mapping?

- Tables in S5. I expect you are making all the assemblies available with the contigs and their annotations, since you're including contig number, position, and variant?

- Please ensure that all sequencing data is available on NCBI GenBank.

Minor points. 

- Suggest being careful with use of "after daptomycin", as it appears that some patients daptomycin use persisted through sampling. This includes some of the specimens with the highest levels of dapto resistance (they were obtained while the patient was on or had just been on daptomycin -- e.g., pt 86, pt 150). 

- There are numerous typos (e.g., line 247 -- I think "150 cluster 1 and 150 cluster 2" is a typo, line 284 looks like it should be 'loss of vancomycin resistance' not 'loss of vancomycin susceptibility'). Suggest careful proofreading.

---

## [Editor Report · Decision Letter 3]

30 Nov 2020

Dear Dr Kinnear,

On behalf of my colleagues and the Academic Editor, Erdal Toprak, I am pleased to inform you that we will be delighted to publish your Research Article in PLOS Biology. 

PRODUCTION PROCESS

Before publication you will see the copyedited word document (within 5 business days) and a PDF proof shortly after that. The copyeditor will be in touch shortly before sending you the copyedited Word document. We will make some revisions at copyediting stage to conform to our general style, and for clarification. When you receive this version you should check and revise it very carefully, including figures, tables, references, and supporting information, because corrections at the next stage (proofs) will be strictly limited to (1) errors in author names or affiliations, (2) errors of scientific fact that would cause misunderstandings to readers, and (3) printer's (introduced) errors. Please return the copyedited file within 2 business days in order to ensure timely delivery of the PDF proof. 

If you are likely to be away when either this document or the proof is sent, please ensure we have contact information of a second person, as we will need you to respond quickly at each point. Given the disruptions resulting from the ongoing COVID-19 pandemic, there may be delays in the production process. We apologise in advance for any inconvenience caused and will do our best to minimize impact as far as possible.

EARLY VERSION

PRESS 

Kind regards,

Erin O'Loughlin

Publishing Editor, 

PLOS Biology

on behalf of

Roland Roberts,

Senior Editor

PLOS Biology